# Flow Synthesis of L-α-Glycerylphosphorylcholine: Studies on Synthetic Routes Applicable to a Flow Reactor and Optimization of Reaction Conditions

**DOI:** 10.3390/pharmaceutics14112480

**Published:** 2022-11-16

**Authors:** Jihun Park, Seungjae Lee, Gyungtak Kim, Yashwardhan R. Malpani, Boyoung Y. Park, Ye-Jin Hwang

**Affiliations:** 1Department of Chemistry and Chemical Engineering, Education and Research Center for Smart Energy and Materials, Inha University, 100 Inha-ro, Michugol-gu, Incheon 22212, Republic of Korea; 2API Synthesis Team, Daewoong-Bio Incorporated, 29-Jeyakdanji-ro, Hyangnam-eup, Hwaseong-si 18608, Republic of Korea; 3College of Pharmacy, Kyung Hee University, 26 Kyughee-daero, Dongdaemun-gu, Seoul 02447, Republic of Korea

**Keywords:** L-α-Glycerylphosphorylcholine, L-α-GPC, flow chemistry, phosphoryl choline, flow reactor

## Abstract

L-α-Glycerylphosphorylcholine (L-α-GPC) has mainly been produced by two methods: extraction from plants rich in phosphatidylcholine and chemical synthesis. However, production through extraction involves difficult processes, such as fermentation, extractions and ripening, and conventional chemical synthesis methods with high-cost reactants and a batch reactor. These methods are not ideal for large-quantity production. Thus, it is important to develop a simple production method of L-α-GPC, which is suitable for mass production without the need for expensive reactants. Here, we studied synthetic L-α-GPC methods that are applicable to a flow synthesis system, which can provide selectivity, reproducibility, scalability, and a high yield in short reaction time using inexpensive starting materials. We developed a two-step synthetic route to produce L-α-GPC, including the synthesis of phosphoryl choline using choline chloride and phosphoryl oxychloride (POCl_3_) as a first step and synthesis of L-α-GPC by reacting phosphoryl choline with (*R*)-(−)-3-chloro-1,2-propanediol (CPD) as a second step under basic conditions. Both steps were separately performed in a customized flow reactor, and reaction conditions were optimized. Finally, phosphoryl choline and L-α-GPC, the products first and second reactions, were successfully synthesized with high conversion yields of 97% and 79%, respectively.

## 1. Introduction

L-α-Glycerylphosphorylcholine (L-α-GPC) is a parasympathetic acetylcholine precursor used to treat Alzheimer’s and dementia and to improve degenerative cognitive ability [1]. The traditional method for preparing L-α-GPC is to isolate and hydrolyze lecithin through biological/chemical process from soybeans and other natural sources, but this method is not ideal for mass production due to its high production costs and low content of GPC and related derivatives [2,3,4]. Several chemical synthetic routes have been developed to make them more suitable for mass production, which are summarized in Figure 1 [5,6,7,8,9,10]. However, the reported synthetic routes still have drawbacks for mass production. One-step epoxide ring-opening reactions using phosphorylcholine (Figure 1a,b) have been reported in high conversion yields (90%) but phosphorylcholine is a relatively expensive reactant that increases the production costs [5,6]. Other reactions, such as Figure 1c,d, are multi-step reactions and require purification procedures (e.g., column chromatography) at each step, resulting in reduced conversion yields [7,8]. Further all these chemical methods suffer from important drawback of the formation of toxic impurities, such as glycidol and glycerol, which require strict control measures to be taken thereafter.

Flow chemistry is an efficient reaction method that provides high conversion yields and high-purity product with reduced reaction times because of its efficient heat and mass transfer [11,12,13,14,15]. In a flow reactor, reaction conditions can be precisely controlled and reproduce uniform reaction conditions after every run, ensuring the high reproducibility of the product quality and strict control of undesired impurities. In addition, the easy scalability of a flow reactor makes it a more suitable reaction method for mass production than conventional batch reactors. However, several challenges remain in the field of flow chemistry, including clog-free reactor design, intermediate purification, and reactor design for multi-step synthesis. A synthetic route of L-α-GPC in a flow reactor has not been reported to the best of our knowledge, and the known synthetic routes are not suitable for flow reactors. This is because the synthetic routes reported for L-α-GPC involve solid precipitation during the reaction or multi-step synthesis. Thus, it is important to develop a simple and high-yield chemical synthesis route for L-α-GPC that can be applied to a flow reactor.

Here, we investigated the chemical synthetic routes applicable to a flow synthesis system. After screening various possible synthetic routes and a careful optimization of reaction conditions in the customized flow reactor, L-α-GPC was successfully synthesized in two-step reactions, including the synthesis of phosphorylcholine using choline chloride and phosphoryl oxychloride (POCl_3_) as the first step, and synthesis of L-α-GPC by reacting phosphoryl choline with (*R*)-(−)-3-chloro-1,2-propanediol (CPD) as a second step, with high conversion yields (95% for the first step and 79% for the second step).

## 2. Materials and Methods

Chemicals: Choline chloride, phosphoryl oxychloride, potassium carbonate, and celite were purchased from Daejung Chemicals & Metals Co., Ltd., Siheung, Republic of Korea. (*R*)-(−)-3-chloro-1,2-propanediol was purchased from TCI. All other chemicals were purchased from Merck.

Flow system: All parts of the system were manually assembled to customize the continuous flow reactor. Chemyx Fusion 200 syringe pumps were used for reactants injection. Among the system components, shut-off valve, static mixing tee, 4-port 3-way flow-switching valves and unions were purchased from Upchurch Scientific. Gastight borosilicate glass syringes (10 mL and 25 mL volumes) and 1/32″ ID tubing (PTFE) were purchased from Runze Fluid. The product-collecting vials located at the end of the system were 50 mL borosilicate vials, purchased from Samwoo Kurex. 

### 2.1. 2-(Trimethylammonio)ethyl Hydrogen Phosphate (Phosphorylcholine) (***3***) Synthesis in a Batch Reactor

Phosphoryl oxychloride (POCl_3_) (5.287 g, 34.48 mmol) was dissolved in 6 mL of anhydrous chloroform in a 50 mL round bottom flask under nitrogen atmosphere and stirred for 10 min at 25 °C. Choline chloride (603 mg, 4.31 mmol) dissolved in deionized water (0.3 mL) was slowly added to POCl_3_ solution. The reaction was carried out at room temperature for 4 h. After the reaction was complete, the reaction mixture was quenched by adding deionized water to remove unreacted POCl_3_. The solution was dried in vacuo at 60 °C and phosphorylcholine was obtained as a colorless liquid (29.42 mmol, 86%). ^1^H NMR (D_2_O, 400 MHz): δ (ppm) 4.04 (m 2H), 3.36 (t, 2H), 2.91 (s, 9H).

### 2.2. 2-(Trimethylammonio)ethyl Hydrogen Phosphate (Phosphorylcholine) (***3***) Synthesis in a Flow Reactor

POCl_3_ (3.965 g, 25.86 mmol) solution dissolved in 4.5 mL of anhydrous chloroform under nitrogen atmosphere and choline chloride (603 mg, 4.31 mmol) solution dissolved in 0.15 mL of deionized water were transferred to gas-tight syringes. Two gas-tight syringes were mounted to the syringe pumps and both solutions at a flow rate of 0.0296 mL/min for POCl_3_ solution, and 0.00165 mL/min for choline chloride solution, and then injected to the flow system. Two streams met at the static mixing tee and entered the reactor (PTFE tube reactor with an internal volume of 30 mL). After 3 h 25 min, product was collected at the vial and quenched with deionized water. The crude product was dried 12 h in vacuo at 60 °C and phosphorylcholine was obtained as a colorless liquid (24.43 mmol, 97.3%). ^1^H NMR (D_2_O, 400 MHz): δ (ppm) 4.04 (m, 2H), 3.36 (t, 2H), 2.91 (s, 9H).

### 2.3. 2-((((R)-2,3-Dihydroxypropoxy)(hydroxy)phosphoryl)oxy)-N,N,N-trimethylethanaminium (L-α-GPC) Synthesis in a Batch Reactor

Phosphocholine chloride calcium salt tetrahydrate (10 g, 30.32 mmol) and potassium carbonate (4.2 g, 30.38 mmol) were dissolved in 10 mL of deionized water and stirred for 1 h at 65 °C. 2 g of celite was poured into the reactor and stirred for 1 h. The reaction mixture was filtered through a vacuum filter and dried in vacuo to yield the phosphocholine chloride potassium salt tetrahydrate (PCK) as a white solid. PCK was dissolved in 8 mL of ethanol and 2 mL of water co-solvent and (*R*)-(−)-3-chloro-1,2-propanediol (CPD) (3.83 g, 34.64 mmol) was added slowly. The reaction was carried out at 75 °C for 24 h. After the reaction was cooled to 25 °C, the solution was dried in vacuo, and L-α-GPC was obtained as a white solid (29.71 mmol, 98%). ^1^H NMR (D_2_O, 400 MHz): δ (ppm) 4.36 (s, 2H), 4.00 (m, 3H), 3.72 (m, 4H), 3.27 (s, 9H).

### 2.4. 2-((((R)-2,3-Dihydroxypropoxy)(hydroxy)phosphoryl)oxy)-N,N,N-trimethylethanaminium (L-α-GPC) Synthesis in a Flow Reactor 

Phosphocholine chloride potassium salt tetrahydrate (PCK) was prepared in the same manner as above. PCK solution in 10 mL of deionized water and (*R*)-(−)-3-chloro-1,2-propanediol (CPD) (3.83 g, 34.64 mmol) were transferred to gas-tight syringes. Two gas-tight syringes were mounted to the syringe pumps and both solutions were injected to the flow system at the same time in a flow rate of 0.0284 mL/min for PCK solution and 0.00484 mL/min for CPD. Two streams met at the static mixing tee and entered the reactor (PTFE tube reactor with an internal volume of 12 mL). After 6 h, product was collected at the vial. The crude product was dried 12 h in vacuo at 60 °C and L-α-GPC was obtained as a white solid (23.6 mmol, 79%). ^1^H NMR (D_2_O, 400 MHz): δ (ppm) 4.36 (s, 2H), 4.00 (m, 3H), 3.72 (m, 4H), 3.27 (s, 9H) (Figure 1), ^13^C-NMR (Methanol-D_4_, 500 MHz) δ (ppm) 54.2–54.3, 59.9–59.9, 63.3, 66.9–67.0, 67.4, 71.9–72.0 (Figure 2), IR spectra (ν, cm^−1^): 1479.1 (CH_2_ Bending), 1215.9 (CN Bending), 1039.4 (PO_4_− Stretching), 966.2 (C-O Stretching) (Figure 3), MS: 258.1 [M+H]+.

### 2.5. Characterization

^1^H nuclear magnetic resonance (NMR) and high-performance liquid chromatography (HPLC) were utilized to confirm the structure and conversion yields of the product in each step. ^1^H NMR spectra of the products solution in D_2_O were observed with a Bruker Avance
III spectrometer at 400 MHz. The scan number was 16. The HPLC analysis was performed on Agilent HPLC instrument equipped with a refractive index detector and a CAPCELL PAK C-18, 5 μm, 4.6 mm × 250 mm column (manufacture: Shisido, Part Number: 90104). The flow rate is 0.5 mL/min. The detector temperature was maintained about 35 °C and the column temperature was maintained at 40 °C. Injection volume was 20 μL and total runtime was 60 min. To prepare samples for analysis, the product was dissolved in deionized (DI) water and filtered by syringe filter with a pore size of 0.45 μm. The eluent was degassed DI water. Before HPLC analysis, the internal column was cleaned by acetonitrile (ACN) and DI water for 1 h each. ACN and DI water for column cleaning were sonicated for 1 h to degas before use. ^13^C NMR spectra of the products solution in Methanol-D_4_ were observed with a Utility Inova instrument at 500 MHz. The infrared (IR) spectra were recorded on a Jasco FT/IR-4100 IR spectrophotometer using the standard bromide pellet preparation method. The mass spectra were recorded on an Agilent 1100 Series mass spectrometer with API-4000 QTRAP detector in positive ionization mode.

## 3. Results and Discussions

Four synthetic routes to prepare L-α-GPC in two steps with inexpensive starting materials, choline chloride (**1**) or glycerol (**2**), were designed and screened to test their applicability in a continuous flow system. In the first two reactions (Figure 2a,b [16,17]), reactant **1** is phosphorylated with POCl_3_ or H_3_PO_4_ to give phosphorylcholine (**3**), and the nucleophilic substitution reaction of **3** with (*R*)-(−)-3-chloro-1,2-propanediol (CPD), which gives L-α-GPC under basic conditions. The other two reactions (Figure 2c,d [18,19]) synthesize glycerol phosphate (**4**) from glycerol (**2**), and the nucleophilic substitution reaction of **4** with chloro choline chloride was expected to give L-α-GPC. Each reaction step was screened individually, first in a batch reactor, and the most promising synthetic route was performed in the customized flow reactor and reaction conditions were optimized. 

Initially, the first steps of all four routes were screened and reaction conditions were optimized by using a batch reactor. The optimized results are summarized in Table 1. Phosphorylation with POCl_3_ and H_3_PO_4_ produces HCl gas and water as by-products. In the H_3_PO_4_ reactions, low conversions (<20%) were observed in initial trials. To enhance the conversion, a Dean–Stark trap was used to continuously remove water by-product. As a result, the phosphorylation of choline chloride (**1**) with H_3_PO_4_ gave a highly enhanced conversion of 80% with a Dean–Stark trap. However, the phosphorylation of glycerol (**2**) with POCl_3_ showed no sign of product formation; only the HPLC peak of glycerol was observed. The highest conversion (86%) was achieved in the choline chloride reaction with POCl_3_. In this reaction, we note that a small amount of water (0.3 mL) was used for the purpose of dissolving choline chloride (**1**), which has limited solubility in CHCl_3_. However, water can deactivate the POCl_3_, so only a small amount has to be used and 0.3 mL (0.0167 mmol) of water was found to be the optimal amount based on POCl_3_ 5.287 g (34.48 mmol) scale. We also believe that water did not significantly affect the reaction outcome because of the high ratio of POCl_3_ used in this reaction scale. Therefore, the phosphorylation of choline chloride with POCl_3_ was selected as the first step of preparing L-α-GPC, and applied in the flow synthesis system. 

The flow reactor for the phosphorylation of choline chloride (**1**) and POCl_3_ was designed, as shown in Figure 4. Each solution of **1** in water and POCl_3_ in chloroform was separately injected using syringe pump and mixed at the mixing tee. The stoichiometry of the reaction was controlled by the flow rate and concentration of the solutions, and the reaction was performed in the PTFE tubular reactor (inner diameter (ID) of 1/32″). Reaction time was controlled by changing the length of the reactor. However, a large difference between the calculated reaction time and the actual reaction time was observed due to the HCl gas by-product generated during the reaction and, accordingly, the length of reactor needed to achieve the target reaction time was experimentally determined. Entire system was purged with nitrogen before use, and the product was collected in the closed collecting chamber. With this set-up, various reaction conditions were screened and optimized for the first step of L-α-GPC synthesis.

Initially, the reaction was performed under similar reaction conditions to the optimized batch reaction (entry 1, Table 2). Unfortunately, conversion of the flow process (60%) was lower than that of the batch process (86%), but the optimum conditions for the flow reactor can be different to those for the batch reactor. This may also be due to the shorter reaction time caused by the HCl gas by-product formation, as mentioned above. To further enhance the conversion for the flow reaction, various reaction parameters were screened, including reaction temperature, molar ratio of POCl_3_ to choline chloride, reaction time, and the amount of H_2_O used to dissolve choline chloride. Table 2 summarizes the reaction conditions and resulting conversions. The experimental details and NMR spectra are available in the Appendix A. 

First, lower or higher reaction temperatures were screened, and a significant temperature effect on reaction time and conversion were found. At a lower temperature (0 °C), HCl gas generation was not observed during the reaction, indicating low reactivity, and only 8% conversion was observed. At higher temperatures (30 °C and 35 °C), a large volume of HCl gas was rapidly generated, and the reaction time was shortened to about 30 min (entry 3 and 4, Table 2). As a result, conversions were much lower than the reaction at 25 °C (32% at 30 °C and 38% at 35 °C). Higher conversions are expected with 2-3 times longer reactors, but these are not practical. Therefore, we screened other reaction conditions to optimize this reaction while the reaction temperature was fixed to 25 °C. 

The molar ratio of POCl_3_ to choline chloride was lowered to 6 and 4. A slower reaction rate and extended reaction time were expected as the POCl_3_ ratio was lowered. At molar ratios of 6 and 4, the reaction times increased by 18 and 46 min, respectively. The optimum POCl_3_ ratio was found to be 6, and the conversion was greatly improved to 88%, which is a comparable value to the 86% conversion found in the batch reaction. At a lower molar ratio of 4, however, the resulting conversion was only 36%, possibly due to the decreased reactivity caused by the decomposition of POCl_3_ in water. For further increased conversion, we varied the amount of water used to dissolve the choline chloride. 

Water was required to dissolve the choline chloride and inject into the flow system without clogging issues. Since water reacts with POCl_3_ to produce phosphoric acid and HCl gas, the amount of water is a critical factor and needs to be optimized. The amount of water was reduced by half (0.15 mL) and the length of the reactor was further extended to provide a longer reaction time. Under these conditions, the molar ratio of water to POCl_3_ is 1:3.1. Although all water reacts with POCl_3_, the molar ratio of reactant (choline chloride) to the remaining POCl_3_ is 1:5.4, indicating that the amount of POCl_3_ is still excessive compared to the reactant. Finally, under this optimized reaction condition, we successfully achieved phosphorylcholine with 97% of conversion. To demonstrate the reproducibility, the first step of the reaction was repeated three times under optimized reaction conditions. As a result, the conversion showed high reproducibility, with few errors (maximum 2%) between runs (Appendix A).

In the second step of L-α-GPC synthesis, the product obtained from the phosphorylation of choline chloride with POCl_3_ was reacted with CPD under the basic condition (Figure 2a). We note that, to find the adequate amount under basic conditions, an ion-exchange process of phosphorylcholine from Ca^2+^ to K^+^ was required prior to use in the second reaction step, and the method for this is described in the experimental section. The second reaction step was first performed in the batch reactor under this reaction condition (entry 1, Table 3). The reaction was performed at pH 10, and KOH was periodically added to maintain the pH during the reaction. Under these reaction conditions, 90% conversion was achieved. However, phosphorylcholine showed limited solubility in ethanol, so this reaction condition could not be applied in a flow reactor. To overcome this problem, we tested a co-solvent system with water. When water was used as a co-solvent, phosphorylcholine was well-dissolved, and the conversion was enhanced by up to 98% (entry 2, Table 3). This second reaction step was applied to a flow reactor using a water/ethanol co-solvent system, and the reaction conditions were optimized.

The second reaction step was performed in a flow reactor designed similarly to the first reaction step (Figure 5). Ion-exchanged phosphorylcholine dissolved in a water or water/ethanol co-solvent and CPD were injected separately by the syringe pumps. The reaction was initially carried out under the same reaction conditions as those used in the batch reactor (75 °C for 24 h with water/ethanol co-solvent). Under these co-solvent reaction conditions, however, the conversion was as low as 28.3%, and phosphorylcholine was observed to precipitate inside the reactor. Therefore, 100% of water was selected as a solvent to fully dissolve phosphorylcholine, and the other reaction conditions were maintained to improve the conversion rate by 50% (Table 4). From the further optimization of reaction time and temperature, we achieved a 79% conversion at 80 °C for 6 h with water as a solvent (Table 5). Three replicate runs under identical reaction conditions demonstrated excellent reproducibility, with low conversion errors between runs (Appendix A).

The most important factor in the second reaction step was found to be the amount of water used to dissolve the phosphorylcholine. As summarized in Table 5, a strong relationship between the amount of water and conversion was observed. This can be understood by looking at the mechanism of the second reaction step (Figure 3 [19]). In the second reaction step, CPD becomes glycidol under basic conditions, and glycidol undergoes a ring-opening reaction with phosphorylcholine to give L-α-GPC [20]. Therefore, the actual pH in real-time has a very critical role in the reaction outcome. In the case of batch reaction, the base was added periodically to maintain the pH, but adding materials to a flow reactor is complicated. We believe that this is the reason for the lower conversion in the flow reaction compared to the batch reaction. In the flow reaction, therefore, the amount of water was controlled to adjust the pH of the reaction mixture, and 10 mL of water was found to be an optimum amount to fully dissolve the phosphorylcholine and provide a basic condition. We note that, with 10 mL of water, the pH of the phosphorylcholine solution was 10.2–10.3.

## 4. Conclusions

In conclusion, we successfully synthesized L-α-GPC in the flow reactor for the first time. We designed a simple two-step synthetic route to produce L-α-GPC that includes phosphorylation with inexpensive choline chloride and epoxide ring-opening reaction with CPD under basic conditions. Each reaction step was modified to adapt to a flow reactor, and the optimization of reaction conditions of the first and second step resulted in high conversions of 97% and 79%, respectively. In this study, we found key reaction parameters that determine the applicability of the flow reactor and the conversion of the reaction. Thus, we considered this result to be meaningful as it not only showed the possibility of producing L-α-GPC in a flow synthesis system, it also showed the key factors for optimization in similar medicinal chemistries.

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
