# Peer review of "Flow Synthesis of L-α-Glycerylphosphorylcholine: Studies on Synthetic Routes Applicable to a Flow Reactor and Optimization of Reaction Conditions"

_pharmaceutics, 2022, doi:10.3390/pharmaceutics14112480_

Round 1

Reviewer 1 Report

The ms is on the flow synthesis of L-α-glycerylphosphorylcholine. Authors should consider the following remarks.

         Scheme 1:

(a): the 2nd step is missing,

(b): the 2nd step is missing,

(c), (d), (e): the 4 or several steps should be marked by “→ → → →”.

-         In the batch and flow syntheses chloroform is not an attractive solvent. Cannot it be replaced by a green one?

-         Compound numbering (e.g. in Scheme 2) is badly needed also in the text.

-         2.1 and 2.2 procedures: the POCl3 is used in a great excess. Its partial hydrolysis is annoying.

-         For the individual steps studied detail, more complete schemes would be necessary.

-         The explanation on page 5, line 177 is not convincing. If the water added will react with some POCl3, the water disappears and will not be able to fulfil its role as a co-solvent for choline chloride.

-         Authors write on page 6, line 215: “At higher temperature (30–35 °C), the rate of deactivation of POCl3 by water dramatically increased than the reaction between choline chloride and POCl3”. How is it possible if only a small amount of water was used (that was assumed to remove the equivalent of POCl3)?

-         The precedents for the appearance of the Ca salt in Scheme 2 (at least for this referee) is missing.

-         Is not it possible to give a more complete spectral characterization? Presently, only 1H NMR data were provided.

-         What may be the productivity of the optimum system?

In summary this is an interesting ms in a good topic. However, the above shortcomings should be clarified during the major revision.

Round 2

Reviewer 1 Report

The au-s tried to reply a few problems, but from the reply not always obvious what, where and how was corrected. The present letter of revision should be completed with another color to be able to see.

The problem of applying the hydrolizable P(O)Cl3 and water together is problematic. It is am unfortunate pair of reactants. This issue should be attended and treated in the ms in a separate paragraph. What is their stoichometry? Why not it causes a problem?

The other problem is that the compound characterization should be included in the main body of ms. It cannot be the decision of au-s to hide it in the Supplementary. Normally all species should be characterized and the spectra should be supplied.

Reviewer 2 Report

Dear authors,

thanks for revise, accept and improve your draft paper. As I said the method is interesting and could be very useful, but, you have to improve your SI file. I can't accept that you can't change the NMR spectra to full scale! Take the time that you need to do that, but it is imperative. The same regarding HRMS and you should show the HPLC data of the pure compound, that you refer you have. 
